# *Ganoderma boninense* Disease of Oil Palm to Significantly Reduce Production After 2050 in Sumatra if Projected Climate Change Occurs

**DOI:** 10.3390/microorganisms7010024

**Published:** 2019-01-19

**Authors:** R Russell M Paterson

**Affiliations:** Department of Plant Protection, Faculty of Agriculture, Universiti Putra Malaysia, 43400 UPM Serdang, Selangor D.E., Malaysia; russellpaterson@upm.edu.my

**Keywords:** global warming, *Elaeis guineensis*, basal stem rot

## Abstract

Palm oil is a valuable crop. This relates to the high economic return from sales of the commodity, where Indonesia is the major producer in the world and the island of Sumatra is the most important region for palm oil production in the country. The island can be considered as a model for other oil palm growing regions in SE Asia. The area in Sumatra with a suitable climate for growing oil palm will decrease in size due to projected climate change as demonstrated specifically herein. The more unsuitable climate will lead to concomitant increases in basal stem rot (BSR) by *Ganoderma boninense*, as previously predicted, which is of major concern to sustainability in SE Asia. A novel approach is described herein, whereby (a) a determination of suitable climate for growing oil palm in Sumatra and (b) deductions to determine future BSR levels on the island were undertaken. The unsuitability of the climate for oil palm is predicted to increase dramatically after 2050 when BSR is predicted to increase to very high levels on most parts of the island. This is likely to make palm oil production unsustainable at some stage between 2050 and 2100. North Sumatra may be more sustainable than the other areas considered in Sumatra. These effects of projected climate change require amelioration before the high levels of BSR and the unsuitable climate for oil palm are realized.

## 1. Introduction

The significance of palm oil is very high [1]. Various components of the commodity are employed in (i) industrial chemicals, (ii) plastics, (iii) cooking, (iv) food, cosmetics, and detergents, and (v) biodiesel. The economy of Indonesia is supported by the economic success of palm oil, as the country is the largest producer, with the island of Sumatra being the most important region. The island is a model for other palm oil growing regions in SE Asia. 

Oil palm is associated with deforestation and conversion of peat soils involved in projected climate change and haze creation [2]. Climate change negatively affects oil palm agronomy, potentially leading to low sustainability [1]. Malaysia is also a large producer and benefits from the advantages and suffers from the disadvantages of the commodity in a similar manner to Indonesia. 

The industry is threatened by various diseases. Basal stem rot (BSR) caused by the fungus *Ganoderma boninense* (Figure 1) has caused severe disease problems for over 80 years [3] and is the most serious disease in Indonesia. BSR is unimportant in other, much lower, palm oil producing regions outside of SE Asia. The current level of disease can be considered as containable, especially compared to the situation if the palms were growing under a suboptimal or unsuitable climate [1,3]. The climate worldwide has changed over this period, which has been discussed frequently by the current author [1,4,5,6] and is generally well known. The fungi delineated by the name *Ganoderma* are variable [7] and will adapt to projected climate change more readily than oil palm by the selection of strains that are more virulent towards oil palm [8], whereas oil palm will be slower to adapt.

The malady has increased since the Second World War [6] and is often reported now in young plants and seedlings, whereas previously, only mature oil palms were infected. The fungus can reduce yields by 50–80% [3]. Over 50% of palms were infected with *Ganoderma* in several areas in the Far East decades ago [9], where the disease was described as very serious. BSR can kill 80% of the stand by the time palms are half way through their economic life span [3]. Furthermore, expansion of industrial oil palm cultivation and BSR disease began early in Sumatra [8], a region where *G. boninense* adaption to the environment is most likely to occur. This region contains the highest levels of disease, implying an association between the length of time of the existence of oil palm and high disease concentrations. The disorder was reported increasingly in inland peninsular Malaysia and Sabah, Malaysia, and in some cases at high levels, whereas before, it had not been detected. BSR was reported increasingly, and at high levels, in oil palm on (i) inland lateritic soils and (ii) peat soils, irrespective of cropping history, when before they were disease-free [10]. By the time of replanting (25 years), 40–50% of palms were lost in some fields, with the majority of standing palms showing disease symptoms [11], whereas levels were low originally. The disease level in Asahan, Indonesia indicated a climate change-related event [12]. This information indicates a trend for increasing BSR with projected climate change. However, the climate for growing oil palm is currently optimal [1,3,13,14] and has been so for many decades. The increase in disease previously reported will be from increased virulence of the fungus, rather than a possible increased susceptibility of oil palm due to a less suitable climate. There are procedures that may assist in reducing the future threats caused by BSR [1] (and see Discussion), although they require testing.

This current paper discusses how projected climate change will affect the BSR of oil palm, oil palm growth, and oil palm sustainability in Sumatra, representing a model of what may happen in SE Asia more generally. 

## 2. Material and Methods

Paterson et al. [13,14] developed models related to suitable climates for growing oil palm under projected climate change. Paterson et al. [13] considered Malaysia and Indonesia, where 85% of oil palms are grown and which effectively constitute the modern industry. Indonesia produces somewhat more than Malaysia. However, Paterson et al. [14] considered the situation for all oil palm growing regions, including SE Asia, Latin America, and Africa, despite their production being only 15% of total when Indonesia and Malaysia are not considered. The current paper focused on Sumatra, which is a large producer within Indonesia, and which was not considered in isolation before in Paterson et al. [13,14] or elsewhere. Figure 2 and Figure 3 provide the maps for Sumatra, which were extracted from Paterson et al. [13,14]. These maps were employed to provide disease assessments through deduction from current disease levels supplied in the literature (Table 1). 

The appearance of basidiomata (Figure 1) was counted on oil palms in specific areas of plantations and compared to oil palms that had no symptoms within the same area. In some cases, core samples were taken and grown on media to provide evidence of the fungus. Full details are provided in the references in Table 1. 

Future suitable climate scenarios for oil palm growth in Indonesia are described fully in Paterson et al. [13,14]. Briefly, the distribution model for oil palm under current and future climate scenarios was developed using CLIMEX for Windows Version 347 (Hearne Scientific Software Pty Ltd., Melbourne, Australia, 2007). Climate data and climate change scenarios were carried out using the CliMond 10´gridded climate data. The potential future climate was characterized using the same five variables based on two global climate models (GCMs), CSIRO-Mk3.053 and MIROC-H (Centre for Climate Research, Japan) with the A1B and A2 Special Report on Emissions Scenarios (SRES) in Reference [13]. These were available as part of the CliMond dataset. CLIMEX parameters were fit using the Global Biodiversity Information Facility, a database of natural history collections from around the world for various species. Information on the global distribution of oil palm was downloaded and used in parameter fitting. A total of 124 records were used. South East Asian distribution data were reserved for validation of the model. 

The oil palm (*Elaeis guineensis* Jacq.) distribution was determined by the Global Biodiversity Information Facility (GBIF) (http://www.gbif.org/, accessed 9 November 2015) and additional literature on the species in CAB Direct (http://www.cabdirect.org/web/about.html, accessed October 2015), and formed the basis for the collection of data on the *E. guineensis* distribution in Reference [14]. A total of 2465 records were used in fitting the parameters. These records may be described as geographically representative of the known distribution of the species. The global study used CSIRO-Mk3·0 and MIROC-H GCM global climate models (GCMs) to model potential future distribution of oil palm. CLIMEX in conjunction with the A2 Special Report on Emissions Scenarios (SRES), a mechanistic niche model using CLIMEX software supports ecological research incorporating the modeling of species’ potential distributions under differing climate scenarios and assumes that climate is the paramount determining factor of plant and poikilothermal animal distributions. CLIMEX output categorized areas according to highly suitable climate (HSC), suitable climate, marginal climate, and unsuitable climate based on other studies through CLIMEX. 

The data were produced in the form of maps [13,14], which were carefully examined to determine the percentage of each climate category in Sumatra. The levels of BSR were assessed given that projected climate change is likely to result in (a) more disease of oil palm [6,13,14,15,16] and (b) highly stressed oil palms [6,13,14,15,16]. The levels of disease were determined by (a) considering how much BSR would increase because of altered climate for oil palm growth [6,13,14,15,16] and (b) attributing a plausible scenario of a 3% increase in BSR every 10 years to take into account the virulence of *G. boninense* per se, as discussed in the Introduction.

## 3. Results and Discussion

The percentages of current disease levels were 37, 39, 51, and 52% for North Sumatra, Sumatra, South Sumatra, and inland Sumatra, respectively (Table 1). Hence, the north may be more sustainable, as it currently has lower disease concentrations. 

Figure 4 indicates the levels of suitable and highly suitable climate together and highly suitable climate alone for growing oil palm in Sumatra. The projected levels of *G. boninense* infection are also provided. The infection will increase significantly because of the decrease in the suitability of the climate for growing oil palm as discussed in References [1,6,13,14,15,16], together with the inherent virulence of *G. boninense*. The suitability of the climate for oil palm growth slightly will increase until 2050, indicative of mountainous regions having an increasingly suitable climate. However, by 2070, there will be a decrease in highly suitable climate, until a very low level is reached by 2100. There will be a fairly small increase in basal stem rot from 41% until 2050, after which time, the increase will be dramatic. The disease level will reach 100% by 2100. 

The situation for Central Sumatra is more serious than in all of Sumatra (Figure 5). The level of disease is higher at the current time and will increase slightly until 2050. The suitability of the climate will decrease at a very high rate for oil palm and highly suitable climate will decrease to a very low level. The amount of disease will increase rapidly to 100% at 2100. The situation in South Sumatra was similar to that in Central Sumatra (Figure 6). The rapid deterioration in suitable climate for oil palm together with the rapid rise in disease will make oil palm unsustainable after 2050 but long before 2100 in most of Sumatra. However, the suitable climate for North Sumatra has a different pattern (Figure 7). There was a low level of highly suitable and suitable climate of ca. 40% probably because of the large number of mountains within this region, with an associated unsuitable climate for growing oil palms. The suitable climate for oil palm will increase linearly until 2050 as the mountainous climate becomes more suitable. The region may become suitable for growing oil palm, but other factors, such as soil type or the slope of the mountains, will affect the ability to grow oil palm. This requires further assessment. The level of disease will increase gradually from 40% to 70% by 2100. Hence, North Sumatra may be more sustainable than the other regions. 

Furthermore, the crop will experience generalized climate stress in Sumatra after 2050, leading to more BSR [1,6,13,14]. Cold stress in current time for growing oil palm in Sumatra was localized largely in mountainous regions, as might be expected, amounting to ca. 10% of Sumatra [13]. This becomes almost absent by 2100, indicating that oil palm could be grown in these regions, possibly with lower BSR. When crops are moved to new regions, the levels of disease can decrease from the “parasites lost” phenomenon [1]. However, other factors will determine if oil palms can be grown in mountainous regions, such as the slope of land and type of soil. There is no heat stress at present: This will become apparent in ca. 5% of Sumatra towards the south east in 2100 [13]. It is likely that oil palm will not grow in that region and, consequently, BSR will be irrelevant. There was no dry stress at present in Sumatra, but data were not available for 2100. The equivalent data in Reference [14] indicated there were none of these stresses in 2050 and 2100, although the definition of the world maps was lower.

Two paths resulting in more disease with projected climate change are recognized herein: (a) An increase in virulence of *G. boninense* and (b) an unsuitable climate for oil palm, making it less resistant to disease. Oil palm in Sumatra will be detrimentally affected by a less suitable climate from 2050 onwards, leading to a high increase in BSR. This indicates a greater risk of unsustainability when the suitable climate and BSR levels are considered. The situation is exacerbated when the virulence of *G. boninense* is considered, with deterioration beginning after 2030 in some cases. The situation in Sumatra acts as a model for other oil palm growing regions in SE Asia. 

Paterson and Lima [1] discussed procedures which may be useful in ameliorating the effect of climate change on oil palms, which will be relevant to controlling BSR and assisting in the sustainability of the crop. For example, the palms could be grown in novel regions outside of SE Asia, where the disease is at low levels, although many of these regions will become unsuitable for growing oil palm under projected climate change. Other factors pertaining to sustainability of the global production of oil palm maybe that the palm could adapt or be adapted by cross-breeding. Genetic manipulation may also be beneficial in the future. The use of arbuscular mycorrhizal fungi with and without reduced tillage, cover crops, biochar, empty fruit bunch application, etc. [1] may all assist in reducing BSR but requires testing. Finally, the situation is urgent, as indicated herein, and proactive steps are required to be taken soon.

## Figures and Tables

**Figure 1 microorganisms-07-00024-f001:**
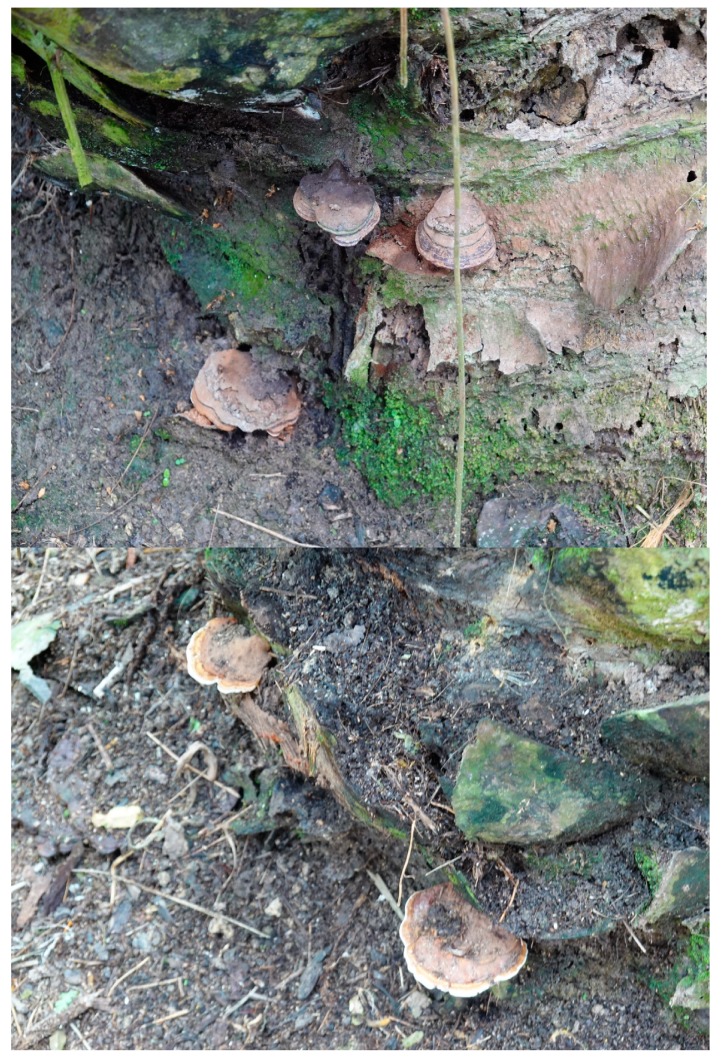
Two images of *Ganoderma boninense* infecting oil palm. The distinctive basidiocarps can be observed clearly (images are from a personal collection of Professor Paterson).

**Figure 2 microorganisms-07-00024-f002:**
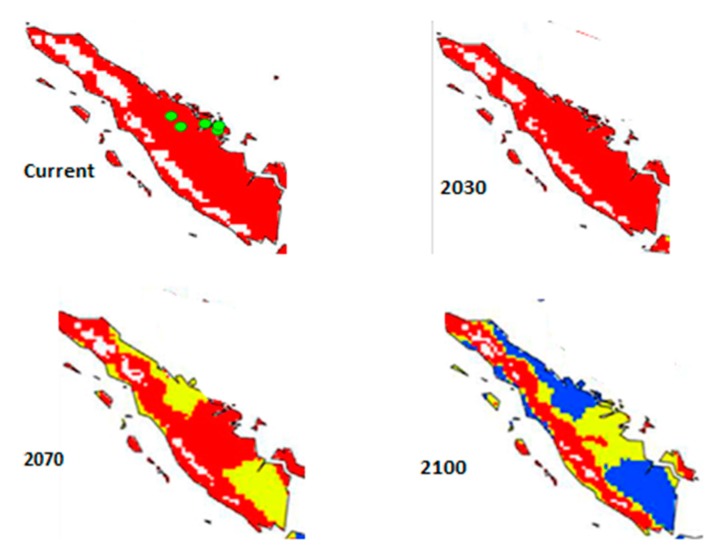
Maps of Sumatra demonstrating suitable climates for growing oil palm at present and in 2030, 2070, and 2100. Red = highly suitable climate; yellow = suitable climate; blue = marginal climate; white = unsuitable climate. The maps represent the CSIRO-Mk3.0 global climate model running the Special Report on Emissions Scenarios (SRES) A1B scenario [13].

**Figure 3 microorganisms-07-00024-f003:**
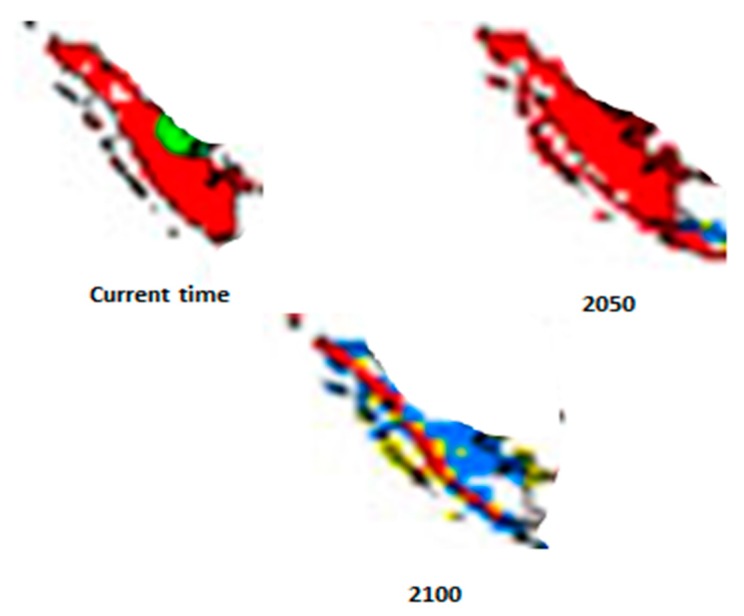
Maps of Sumatra demonstrating suitable climates for growing oil palm at present and in 2050 and 2100. Red = highly suitable climate; yellow = suitable climate; blue = marginal climate; white = unsuitable climate. (Green is the location of an existing plantation). The maps represent the CSIRO-Mk3·0 running the SRES A2 scenario [14].

**Figure 4 microorganisms-07-00024-f004:**
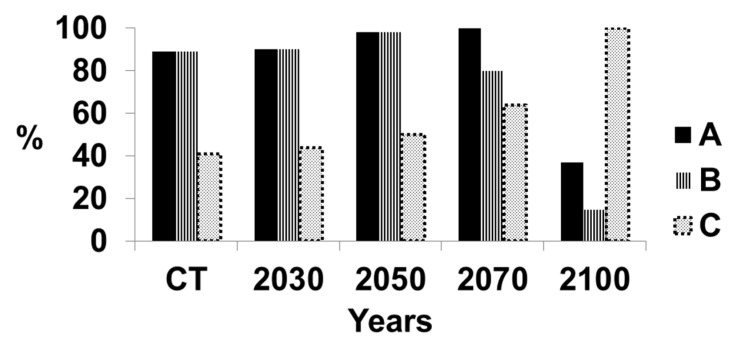
Suitable climate for growing oil palm and percentage of infection by *Ganoderma boninense* in the whole of Sumatra, Indonesia. A = Percentage area of Sumatra with a highly suitable climate and suitable climate for growing oil palm; the remainder of Sumatra has an either marginal or unsuitable climate for growing oil palm. B = Percentage area of Sumatra with a highly suitable climate only for growing oil palm. C = Percentage of oil palms with basal stem rot.

**Figure 5 microorganisms-07-00024-f005:**
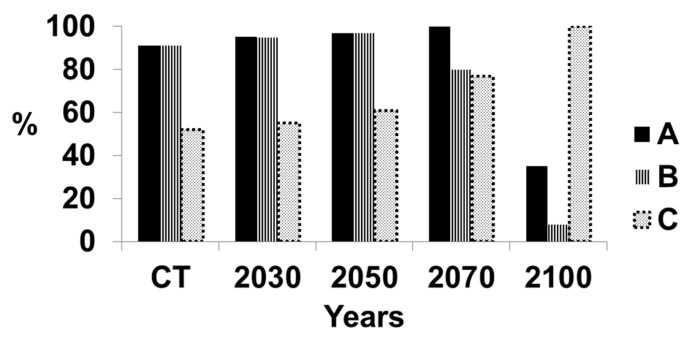
Suitable climate for growing oil palm and percentage of infection by *Ganoderma boninense* in inland Sumatra, Indonesia. A = Percentage area of Sumatra with a highly suitable climate and suitable climate for growing oil palm; the remainder of Sumatra has an either marginal or unsuitable climate for growing oil palm. B = Percentage area of Sumatra with a highly suitable climate only for growing oil palm. C = Percentage of oil palms with basal stem rot.

**Figure 6 microorganisms-07-00024-f006:**
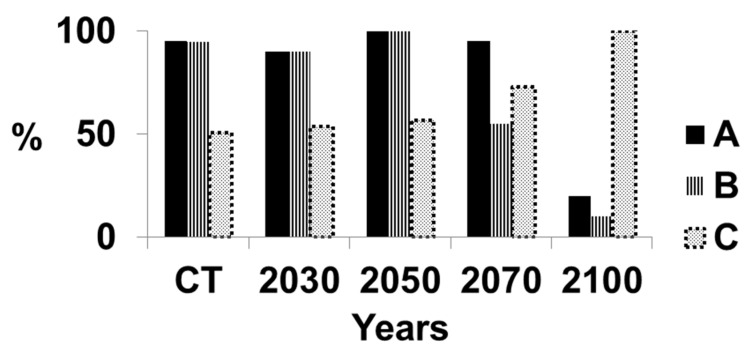
Suitable climate for growing oil palm and percentage of infection by *Ganoderma boninense* in south Sumatra, Indonesia. A = Percentage area of Sumatra with a highly suitable climate and suitable climate for growing oil palm; the remainder of Sumatra has an either marginal or unsuitable climate for growing oil palm. B = Percentage area of Sumatra with a highly suitable climate only for growing oil palm. C = Percentage of oil palms with basal stem rot.

**Figure 7 microorganisms-07-00024-f007:**
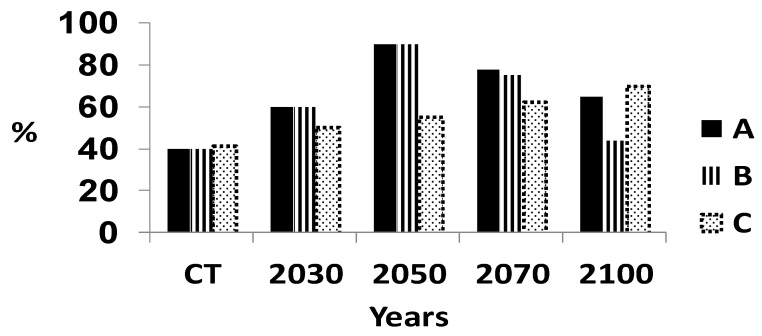
Suitable climate for growing oil palm and percentage of infection by *Ganoderma boninense* in north Sumatra, Indonesia. A = Percentage area of Sumatra with a highly suitable climate and suitable climate for growing oil palm; the remainder of Sumatra has an either marginal or unsuitable climate for growing oil palm. B = Percentage area of Sumatra with a highly suitable climate only for growing oil palm. C = Percentage of oil palms with basal stem rot.

**Table 1 microorganisms-07-00024-t001:** Infection levels of basal stem rot by *Ganoderma boninense* in Sumatra, Indonesia.

Region	Infection Level (%)	Citation
All Sumatra	39	[3,10]
Inland Sumatra	52	[3,10]
North Sumatra	37	[12,17,18]
South Sumatra	51	[19]

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
