# Peer review of "Ganoderma boninense Disease of Oil Palm to Significantly Reduce Production After 2050 in Sumatra if Projected Climate Change Occurs"

_microorganisms, 2019, doi:10.3390/microorganisms7010024_

Round 1
Reviewer 1 Report
General.
The manuscript as it stands lacks quantitative information. The results of the modelling using CLIMEX has presumably produced predictions of the percent suitable climate and basal stem rot that can be placed on graphs in a more informative way than they are in Figures 1–4 as currently presented. The Figures as they stand have very peculiar scales. The year shouldn’t start at ‘0’; is it meant to be 2010? The scale is uneven, having steps of 20 years changing to 30 years at years 2070–2100. There are no gridlines on the graph to help the reader read off quantitative values. The legends “Both suitable climates, Only high suitable, Only high suitability, High suitable climate” are very confusing and need to be improved. It is not clear what is meant by “Both suitable climates”. Do you mean “suitable both for growing oil palm and for Ganoderma boninense infection to occur”? If so, it can be more clearly stated by deleting the legend and putting that information in the figure caption. The legend “Only high suitable” and its other variants are particularly confusing. Only highly suitable for what? Growing oil palm or for infection to occur? Again, deleting the legends in their entirety and adding explanations in the figure captions would be preferable. In the absence of accessible quantitative information, this manuscript adds very little to the knowledge about the future suitability for growing oil palm beyond which is currently available from the myriad of other papers that the author has published.
Specific.
Line 20. Perhaps “at some stage between 2050 and 2100” would be better than “after 2050 but before 2100”.
Line 44. Replace “palm” by “palms”.
Line 62. Insert “how” before “climate”.
Lines 72–73. Why are “Data” and “Scenarios” in capital letters?
Line 80. Replace “was” by “were”.
Line 81. Start sentence with “The oil palm (Elaeis guineensis Jacq.) distribution …”
Line 84. Insert “the” after “on”.
Line 104. It is implied here that Figure 1 has two parts, (a) and (b), but in reality Figure 1 has only one part.
Line 106. Insert a comma after “2050”.
Line 112. Insert “the” after “higher at”.
Line 112. Replace “remains fairly constant” by “increases slightly”.
Line 114. Insert a comma after “rapidly”.
Lines 125–126. The climate stress to the crop is not well quantified. The sentence as it stands is very vague.
Line 156. Insert that it is the Fifth Edition of the book that is being cited.
Line 190. “In” not “in”, and “controlling” not “controllin”.
Lines 196–198. No year is given for this reference.
Author Response
Reviewer 1
General.
The manuscript as it stands lacks quantitative information.
Accurate quantitative information of climate change effects in the future is not really possible I am afraid. We cannot accurately predict the future. What can be done by models and by experienced researchers of climate change is provide plausible scenarios. When providing scenarios for basal stem rot under climate change the current author has considerable experience of the disease and on climate change effects in the agricultural field and so has authority.
The results of the modelling using CLIMEX has presumably produced predictions of the percent suitable climate and basal stem rot that can be placed on graphs in a more informative way than they are in Figures 1–4 as currently presented.
Well, without wishing to be pedantic, the modelling data are not predictions but plausible scenarios. This concept is accepted generally for this type of modelling. CLIMEX gave scenarios of suitable climate for oil palm. From this and the current levels of current basal stem rot, a plausible level of basal stem rot in the future was deduced on the basis of my informed experience. I agree the figures need to be better presented as dealt with below. And I added some more figures to make the paper easier to relate to (see revised paper).
The Figures as they stand have very peculiar scales. The year shouldn’t start at ‘0’; is it meant to be 2010? The scale is uneven, having steps of 20 years changing to 30 years at years 2070–2100. There are no gridlines on the graph to help the reader read off quantitative values.
The reviewer is quite right. I agree starting at “0” is odd and I should have qualified what this meant. I have changed “0” to “current time (CT)”, the term used in the papers Paterson et al. (2015; 2017) on which parts of the figures are based. Also, the reviewer is correct about the scale. They should not be seen as connected as I have portrayed. Rather they are discrete points in time. I have changed them to bar graphs for each time period. I thank the reviewer for these comments, well taken.
The legends “Both suitable climates, Only high suitable, Only high suitability, High suitable climate” are very confusing and need to be improved. It is not clear what is meant by “Both suitable climates”. Do you mean “suitable both for growing oil palm and for Ganoderma boninense infection to occur”? If so, it can be more clearly stated by deleting the legend and putting that information in the figure caption. The legend “Only high suitable” and its other variants are particularly confusing. Only highly suitable for what? Growing oil palm or for infection to occur? Again, deleting the legends in their entirety and adding explanations in the figure captions would be preferable.
The reviewer is correct here too. I have addressed these points now in the revised manuscript.
In the absence of accessible quantitative information, this manuscript adds very little to the knowledge about the future suitability for growing oil palm beyond which is currently available from the myriad of other papers that the author has published.
The current information on basal stem rot in the future is the first such report. It is not available in the 2 papers I published on suitable climate for oil palm growth or any other of my papers on oil palm. Over a period of ca. 10 years I have published ca. 9 papers related to oil palm. None contain this information on basal stem rot. It is not a “myriad” of papers, with all due respect. However, the title of my current paper was confusing which implied that oil palm growth information was new data. This is now changed.
Specific.
Line 20. Perhaps “at some stage between 2050 and 2100” would be better than “after 2050 but before 2100”.
Done.
Line 44. Replace “palm” by “palms”.
Done
Line 62. Insert “how” before “climate”.
Done
Lines 72–73. Why are “Data” and “Scenarios” in capital letters?
Corrected
Line 80. Replace “was” by “were”.
Done
Line 81. Start sentence with “The oil palm (Elaeis guineensis Jacq.) distribution …”
Done
Line 84. Insert “the” after “on”.
Done
Line 104. It is implied here that Figure 1 has two parts, (a) and (b), but in reality Figure 1 has only one part.
Figures have been changed and this does not now occur.
Line 106. Insert a comma after “2050”.
Done
Line 112. Insert “the” after “higher at”.
Done
Line 112. Replace “remains fairly constant” by “increases slightly”.
Done
Line 114. Insert a comma after “rapidly”.
Done
Lines 125–126. The climate stress to the crop is not well quantified. The sentence as it stands is very vague.
I have discussed this much further and after comment of other reviewer.
Line 156. Insert that it is the Fifth Edition of the book that is being cited.
Done
Line 190. “In” not “in”, and “controlling” not “controllin”.
Done
Lines 196–198. No year is given for this reference.
2000 added
TITLE
Could simplify: E.g. Ganoderma boninense disease of oil palm is expected to significantly reduce production in Sumatra after 2050 if projected climate changes occur.
GENERAL
This short paper appears to present most of the work already published by the author and others. What are the additional findings? I ask because few results are given here - data is scant. The Materials and Methods are insufficiently detailed. No statistical analysis is provided. The writing often appears subjective. I have learnt your opinions rather than gained a scientific understanding of how you arrived at them - the details on methods are too vague.
Climate change is referred to frequently. It might appear more objective if you referred to 'projected climate changes' and if you considered, even briefly, other factors pertaining to the future sustainability of the global production of oil palm. E.g. (1) Oil palm can adapt or be adapted by cross breeding, use of endophytic bacteria, genetic manipulation etc. But none of these plant breeding or any other tools are mentioned here or reviewed. (2) If the projected changes were studied globally, presumably, presently unsuitable areas for production may become suitable. In an International Journal we expect authors to use the global models rather than take a parochial view of production - a view that might be better suited to a regional forum. Can global production be maintained/increased by utilizing ‘new’ regions (in the tropics - SE Asia/Africa/Central America - or slightly more temperate latitudes) where projected CC may be favourable for the future OP production?
The word “very” is a subjective, meaningless word that surely has no place in scientific literature? I noticed it on lines 26, 46 and twice on line 113.
ABSTRACT
Condense opening sentences. E.g. PO is a valuable crop. The abstract is purely subjective; could you mention yield/yield decreases in quantitative terms here?
Line 14 Clarify E.g. ‘…will decrease if CC projections are accurate and BSR increases’
L15 There is surely nothing novel about a survey of literature. It is not appear an appropriate term anyway for a short paper citing but 19 references – 13 if we exclude the author’s.
INTRODUCTION
L42. The introduction/Discussion should consider production factors, other than climate, that may improve adaptation (see above suggestions).
L62 suggest insert “how projected” after ‘discusses’. Re using letters, (a), (b), (c). These are not really necessary for enumeration here, in this context.
MATERIALS AND METHODS
L66 Simplify: E.g. Paterson et al developed models relating OP growth to climate
L67 This sentence is meaningless. We need to be told How it was done with specific details
Para 2 Where in this paper is information on the climate favoured by BSR? And, what are the projected Climate details for Sumatra? Summarise here if you want the reader to appreciate by gaining an understanding. A regional audience may understand. For the wider readership you require too much knowledge that I lack.
L96 (a) references are needed to support statement – as you have for (b) with 5 references
L97 (a) We need to know HOW this “consideration” was determined?
L98 (b) No references are given to support this statement. As it is, this ‘3%’ appears to be a subjective guess.
RESULTS
L101-2 Have you any comment as to how ‘variation in seasonal conditions’ may influence this?
L102 Reference here to “all Sumatra” is inappropriate - and confusing.
L104 Please clarify this sentence. It does not make sense
L105 But, HOW were the projected levels of BSR arrived at? Such detail is essential for the paper‘s credibility.
L1235 What is this “climate stress”? Can you specify as drought/water-logging/temperature? Presumably you could be more specific. That would be useful.
REFERENCES
References 2, 7 and 8 need to acknowledge all the authors.
Author Response
Reviewer 2
Comments and Suggestions for Authors
TITLE
Could simplify: E.g. Ganoderma boninense disease of oil palm is expected to significantly reduce production in Sumatra after 2050 if projected climate changes occur.
Thank you for the improved title which I have largely employed.
GENERAL
This short paper appears to present most of the work already published by the author and others. What are the additional findings? I ask because few results are given here - data is scant.
The basal stem rot scenarios in the future are new, whilst using the published work on oil palm suitability in an original manner to cover Sumatra. Sumatra provides a model for the ability to assess basal stem rot scenarios with projected climate change which could be applied more widely to other regions. This information is unavailable otherwise. This short paper is bigger now with added Figures and text to increase comprehension and interest to a microbiological (or mycological) audience.
The Materials and Methods are insufficiently detailed. No statistical analysis is provided. The writing often appears subjective. I have learnt your opinions rather than gained a scientific understanding of how you arrived at them - the details on methods are too vague.
I have improved the Material and Methods by providing climate suitability maps of Sumatra as figures to make clearer how this information was obtained. I feel I have provided enough abbreviated information in the paper and by providing the references (Paterson et al. 2015; 2017) to where more detail can be obtained. This information indicates the complex mathematical procedures that are involved in this work. The deduced basal stem rot levels are plausible scenarios (often used in modelling) demonstrating the problem that may arise.
Climate change is referred to frequently. It might appear more objective if you referred to 'projected climate changes'
I have added “projected” to “climate change” in almost all places throughout.
and if you considered, even briefly, other factors pertaining to the future sustainability of the global production of oil palm. E.g. (1) Oil palm can adapt or be adapted by cross breeding, use of endophytic bacteria, genetic manipulation etc. But none of these plant breeding or any other tools are mentioned here or reviewed.
Thank you for this. The last sentence does indicate that amelioration is required. However, I added more information based on (Paterson and Lima 2018).
These to be added…..
(2) If the projected changes were studied globally, presumably, presently unsuitable areas for production may become suitable. In an International Journal we expect authors to use the global models rather than take a parochial view of production - a view that might be better suited to a regional forum.
Well, I am not discussing oil palm production or growth per se as the main thrust of the paper. Only as a way of assessing basal stem rot in the future. Admittedly, the title of the paper was confusing in this respect which has now been changed as you suggested.
Basal stem rot is not a global concern in the sense it is not significant outside SE Asia. It is the major oil palm disease only of Indonesia and Malaysia. Indonesia and Malaysia account for 85% of palm oil production and Indonesia produces more than Malaysia. In a sense the global situation is Indonesia and Malaysia. Sumatra is one of the largest producers of these two countries and so is highly significant. Hence I am justified in discussing Sumatra only and as a model for SE Asia.
Can global production be maintained/increased by utilizing ‘new’ regions (in the tropics - SE Asia/Africa/Central America - or slightly more temperate latitudes) where projected CC may be favourable for the future OP production?
This is an interesting question. I discuss this in two of my previous papers in relation to growing oil palm (Paterson et al 2017; 2018) although not in relation to basal stem rot. Indeed there are some areas that become favourable for growing oil palm in terms of climate. Using these areas to grow oil palm to prevent basal stem rot makes sense. I now discuss this in the paper. Thank you.
The word “very” is a subjective, meaningless word that surely has no place in scientific literature? I noticed it on lines 26, 46 and twice on line 113.
Deleted.
ABSTRACT
Condense opening sentences. E.g. PO is a valuable crop.
Done.
The abstract is purely subjective; could you mention yield/yield decreases in quantitative terms here?
Well, there are some data that relates to the main text here regarding BSR increasing as suitable oil palm climate decreases. In fact there is no information on yield under future climate change and the paper is not about yield per se.
Line 14 Clarify E.g. ‘…will decrease if CC projections are accurate and BSR increases’
I have made this more meaningful.
L15 There is surely nothing novel about a survey of literature. It is not appear an appropriate term anyway for a short paper citing but 19 references – 13 if we exclude the author’s.
Yes, correct. I have amended this sentence.
INTRODUCTION
L42. The introduction/Discussion should consider production factors, other than climate, that may improve adaptation (see above suggestions).
I have used a form of words in the Introduction to address this and included detail in the Discussion.
L62 suggest insert “how projected” after ‘discusses’. Re using letters, (a), (b), (c). These are not really necessary for enumeration here, in this context.
Done
MATERIALS AND METHODS
L66 Simplify: E.g. Paterson et al developed models relating OP growth to climate
Done
L67 This sentence is meaningless. We need to be told How it was done with specific details
Information now added.
Para 2 Where in this paper is information on the climate favoured by BSR?
The climate favoured by BSR is discussed in the Introduction which is quite a large part of the text.
And, what are the projected Climate details for Sumatra?
I have now provided two figures to give the projected climates.
Summarise here if you want the reader to appreciate by gaining an understanding. A regional audience may understand. For the wider readership you require too much knowledge that I lack
.
I have summarised this information in the first paragraph of the Material and Methods and a small amount in the Introduction. Although it appears regional because I am discussing only Sumatra, it constitutes a large proportion of palm oil production, because Indonesia is such a major player. You also need to know that BSR is very nearly confined to Indonesia, Malaysia and the rest of SE Asia. I hope this is clear now.
L96 (a) references are needed to support statement – as you have for (b) with 5 references
All 5 references apply to (a) too. I was trying to be succinct by placing the references at the end of the sentence but I agree placing them in the two places makes it clearer.
L97 (a) We need to know HOW this “consideration” was determined?
I have altered the text on this point, and see next point.
L98 (b) No references are given to support this statement. As it is, this ‘3%’ appears to be a subjective guess.
The 3% provides a plausible scenario based on my experience from working with BSR. Even discussion about future climate change using complex models, for example, give scenarios and are not meant as accurate predictions. The value is a plausible scenario to take into account the virulence of G. boninense per se as discussed in the Introduction.
RESULTS
L101-2 Have you any comment as to how ‘variation in seasonal conditions’ may influence this?
I do not have any particular comment on seasonal conditions which I trust is not a particular problem for the reviewer. I am considering a larger scale than seasons as interesting as these would be.
L102 Reference here to “all Sumatra” is inappropriate - and confusing.
I changed the sentence.
L104 Please clarify this sentence. It does not make sense
Done, I have changed the Results considerably partly too from comments of the other reviewer.
L105 But, HOW were the projected levels of BSR arrived at? Such detail is essential for the paper‘s credibility.
I have added a sentence with references to clarify this and the information is provided in the Material and Methods now.
L1235 What is this “climate stress”? Can you specify as drought/water-logging/temperature? Presumably you could be more specific. That would be useful.
Thank you for this. I have added a paragraph of useful information based on your comment.
REFERENCES
References 2, 7 and 8 need to acknowledge all the authors.
I included first 10 authors in 2 cases as permitted. Otherwise done.
Round 2
Reviewer 1 Report
In Figures 4 to 7, I suggest changing "highly suitable climate plus suitable climate"to "suitable or highly suitable climate". Also, after the semicolon, change the rest of the sentence to "the remainder of Sumatra was either marginal or unsuitable climate for growing oil palm."
Author Response
Comments and Suggestions for Authors
In Figures 4 to 7, I suggest changing "highly suitable climate plus suitable climate"to "suitable or highly suitable climate".
I have changed to “highly suitable climate and suitable climate”.
Also, after the semicolon, change the rest of the sentence to "the remainder of Sumatra was either marginal or unsuitable climate for growing oil palm."
Done
Thank you
Reviewer 2 Report
L 173-4 I suggest the acceptability of GM will increase considerably in the decades ahead and that it is better not to qualify your comment about their potential role?
Fig 4 and others: I have some difficulty understanding the distinction between A and B, and indeed why the two categories are needed. It would be helpful if that could be made clearer.
Author Response
Comments and Suggestions for Authors
L 173-4 I suggest the acceptability of GM will increase considerably in the decades ahead and that it is better not to qualify your comment about their potential role?
Agreed and so deleted.
Fig 4 and others: I have some difficulty understanding the distinction between A and B, and indeed why the two categories are needed. It would be helpful if that could be made clearer.
I can see the point so I have simply deleted the allusion to the differences in Figure 4. The fact that it is from a different paper that is cited and uses different criteria to provide different scenarios is enough information upon reflection.
Thank you.